# Detection of Synthetic Antioxidants: What Factors Affect the Efficiency in the Chromatographic Analysis and in the Electrochemical Analysis?

**DOI:** 10.3390/molecules27207137

**Published:** 2022-10-21

**Authors:** Danielle Gonçalves-Filho, Djenaine De Souza

**Affiliations:** Laboratory of Electroanalytical Applied to Biotechnology and Food Engineering (LEABE), Chemistry Institute, Uberlândia Federal University, Major Jerônimo Street, 566, Patos de Minas 38700-002, MG, Brazil

**Keywords:** food additives, synthetic antioxidants, butylated hydroxytoluene, chromatographic analysis, voltammetric analysis, foodstuffs

## Abstract

Antioxidants are food additives largely employed to inhibit oxidative reactions in foodstuffs rich in oils and fat lipids, extending the shelf life of foodstuffs and inhibiting alterations in color, flavor, smell, and loss of nutritional value. However, various research has demonstrated that the inadequate use of synthetic antioxidants results in environmental and health problems due to the fact that some of these compounds present toxicity, and their presence in the human body, in high concentrations, is related to the development of some cancer types and other diseases. Therefore, the development of analytical methods for identifying and quantifying synthetic antioxidants in foodstuffs is fundamental to quality control and in ensuring consumer food safety. This review describes the recent chromatographic and electrochemical techniques used in the detection of synthetic phenolic antioxidants in foodstuffs, highlighting the main characteristics, advantages and disadvantages of these methods, and specific typical features, which include extraction methods for sample preparation and materials used in the working electrode construction, considering chromatographic and voltammetric methods, since these specific features influence the efficiency in the analysis.

## 1. General Considerations

In the last decades, the worldwide population has undergone intense changes in its lifestyle, promoted mainly due to technological advances that have brought greater comfort, convenience, and general quality of life and allowed a considerable increase in the life expectations of the population. These alterations in lifestyle have made food industries undergo significant changes and/or adequacy in the composition of industrialized products to meet new needs that have emerged, from adjustments to legislation to nutritional deficiencies and technological attributes. Moreover, the growth in the number of consumers requires a large-scale production of foodstuffs, which offer tasty products, are safe from physicochemical and microbiological points of view, are easy to prepare, and meet the nutritional demands of a large part of the population [1,2].

All these aspects have promoted changes in industrial processes, allowing for an improvement and increase in the quantities and types of industrialized foods and the production of a wide variety of foodstuffs developed considering different peculiarities related to sex, age, and cultural aspects. However, the modern industrial production of foods has required the employ of some chemical compounds during the processing, called food additives and nutritional additives. The employ of food additives results in improvements of the physicochemical (homogeneity, stability, pH, and viscosity), sensory (color, flavor, and smell), and microbiological properties (prevention or delay of food spoilage due to the growth of microorganisms) of processed foods, and extends the shelf life of food, ensuring food safety to consumers. The use of nutritional additives already results in the restoration, fortification, and/or enrichment of the nutritional properties of foodstuffs [2,3,4].

According to the Codex Alimentarius [5], there are about 230 compounds classified according to their functionalities into 25 classes of food additives that are regulated by specific legislation of each country following a rigorous policy of food safety [4,6]. Among food additives, synthetic phenolic antioxidants are used to prolong the shelf life of foodstuffs due to the inhibition of the aging processes, such as the oxidation of lipid compounds that promote alteration in color and loss of nutritional value. However, synthetic antioxidants should be used appropriately, considering the type of compounds, the limited amount allowed by legislation, and the type of food production, due to the fact that their inefficient use can result in adverse health effects, such as allergic and idiosyncratic reactions in consumers, and change the quality of foodstuffs. Furthermore, some research has shown that the use of inappropriate amounts of these substances has genotoxic effects, demanding rigorous control of the type and quantity of food additives in foodstuffs.

According to supervisory organs of the food industry, the identification and quantification of synthetic phenolic antioxidants in foodstuffs should be performed using previous separation by chromatographic techniques with the steps of pretreatment and cleanup of samples, which are based on liquid–liquid extraction (LLE) [7,8,9], solid-phase extraction (SPE) [10,11,12,13], and the combination of other extraction methods [14,15,16,17,18,19,20]. However, in the last decades, some researchers have demonstrated that the use of voltammetric techniques in the identification and quantification of antioxidants in foodstuffs allows some advantages, such as low cost of instrumentation and inputs used, reduced analysis time and requirements of few and simplified sample preparation steps when compared with chromatographic analysis since voltammetric analysis can be performed in colored materials and samples with dispersed solid particles [21]. The use of voltammetric techniques with potential pulse applications, such as square wave voltammetry, results in sensitivity similar to those obtained using chromatographic techniques [22].

In practice, the success of the use of chromatographic separation, followed by detection using different detectors, requires adequate performance of extraction methods, while the success of the employ of voltammetric techniques is related to the materials used in the preparation of the working electrode and the interface on which the reaction of interest occurs. The main steps in synthetic phenolic antioxidants and other food additives and their detection are shown in Figure 1.

The performance of chromatographic and voltammetric analysis of antioxidants is evaluated by sensitivity, selectivity, precision, accuracy, linear range, and robustness parameters, which are obtained from an adequate choice of pretreatment steps of the samples and of the materials of the working electrode, considering chromatographic and voltammetric analysis. Therefore, this review shows recent reports on the detection of antioxidants in foodstuffs using chromatographic and electrochemical techniques, indicating the advantages, disadvantages, requirements, and types of preparation and purification steps of the samples of foodstuffs before chemical analysis or the types of working electrodes employed.

## 2. Antioxidants

Foods, from natural sources or industrially processed, rich in oils and fat lipids, can suffer oxidative reactions, which results mainly in undesirable flavor and smell and reduction in nutritional values, among other phenomena [4,23]. These oxidative reactions occur due to the presence of specific enzymes or molecular oxygen, highly reactive species that react with fatty acid double bonds or with oxygen, nitrogen, or sulfur elements present in the chemical structure of compounds contained in foodstuffs. These reactions produce reactive free radicals that promote chain reactions that impair the organoleptic properties (color, smell, taste, and texture) and technological properties (homogeneity, pH, stability, and viscosity) and reduce the nutritional value and mainly the shelf life of foodstuffs [1,24,25].

Antioxidants are an important class of food additives, largely employed to inhibit oxidative reactions and extend foodstuff shelf life, without losses in the properties mentioned above [26]. They are effective against oxidation and polymerization reactions but do not act on the hydrolysis and reversal of fatty acids, because they are additives that prevent oxidative processes from occurring but are not able to reverse them if the process has already occurred [25,27].

### 2.1. Classification of Antioxidants

Several compounds, from natural and synthetic sources, have presented antioxidant properties [26]. In the past, natural antioxidants, extracted from plants and animals, were intensively used, but in the last decades, they were replaced by synthetic substances, which have a lower cost and a higher level of purity and present more uniform antioxidant properties [28,29]. Among synthetic antioxidants, phenolic compounds are a family of high-efficiency lipophilic antioxidants, when compared with natural antioxidants, such as tocopherols.

Synthetic phenolic antioxidants have a stable chemical structure, high thermal stability, and strong antioxidant capacity and can be produced in large quantities. The main synthetic phenolic antioxidants are 2 and 3-tert-butyl-4-hydroxyanisole (BHA), 2,6-di-tert-butyl-4-methylphenol or butylated hydroxytoluene (BHT), tert-butyl hydroquinone (TBHQ), gallate antioxidant, propyl gallate (PG), and octyl gallate (OG). Due to their high stability and small effect on food color and flavor, the use of synthetic phenolic antioxidants in foodstuff production has increased year by year [2,29,30]. The chemical formula and structure of these antioxidants are shown in Table 1, with some physicochemical properties and information about their toxicity.

Antioxidant activity depends on many factors, such as lipid composition, antioxidant concentration, temperature, oxygen pressure, the presence of other antioxidants, water activity, and other common foodstuff components, such as protein. The use of antioxidants during industrial production of foodstuffs is only authorized if there is a reasonable technological need, which cannot be met by other technically feasible methods, and the criteria for the choice of antioxidants are based on the type of lipid that requires antioxidant protection, the physical state of foodstuffs, the storage management, and water activity.

### 2.2. Toxicity of Antioxidants

In the last decades, the intensified use of some synthetic antioxidants allowed for their presence in various environmental matrices, such as water, soil and sediments, petroleum products, urban dust, and biological samples, such as blood, urine, and tissues. In recent years, many studies, in vitro and in vivo, have investigated the antioxidant toxicity and the relationship between the consumption of foodstuffs rich in synthetic antioxidants and some cancer types and other diseases [8,9,24,29,31,32,33].

Therefore, as the consumption of industrialized foods cannot put a risk to the health of consumers, well-developed food safety policies were established by the Food and Agriculture Organization (FAO), the World Health Organization (WHO), the European Union, the main world foodstuff regulatory agencies, through the implementation of the *Codex Alimentarius* [34], which established the conditions under which all food additives can be used in the composition of processed foods. In Brazil, the National Sanitary Vigilance Agency (ANVISA) is the authority responsible for the authorization and supervision of the types of additives and their respective quantities in foodstuffs [35]. In addition to following national guidelines, food-exporting industries need to comply with international quality standards, specific regulations, and legislation of each consumer country and adequately meet consumer expectations in an adequate quantity of food additives, including antioxidants [4].

The food safety policies proposed by the Codex Committee on Food Additives and Contaminants (CCFAC) are the benchmark for food industries, regulatory bodies, and authorities in the area of food safety around the world, resulting in the creation of specific legislation in other countries [34,36]. The CCFAC provided the general guidelines on the acceptable daily intake (ADI) of antioxidants, which is related to the maximum amount of an antioxidant that can be ingested daily over a lifetime with no appreciable health risk [30,37]. ADI is expressed in milligrams of the antioxidant per kilogram of the body per day (mg/kg/day) and considers the lowest no observed adverse effect level (NOAEL) determined from long-term animal (in vivo) studies. ADI for antioxidants is shown in Table 1.

Despite the necessity and the several advantages promoted by using phenolic antioxidants in foodstuff production, their toxicological effects have been the subject of several scientific research and much controversy in recent years. According to the Food and Drug Administration (FDA) [38], BHT and BHA are considered carcinogenic compounds, and for the safety of consumers, their use is limited from 50 parts per billion for dehydrated and dry foods to 200 parts per billion in emulsions, alone or in combination. However, in some countries, such as Japan, Sweden, Australia, and some European countries, BHT is banned. According to the FDA, gallates present some toxicological effects, including dermatological allergies, and propyl gallate presents carcinogenic effects.

The International Agency for Research on Cancer (IARC) [39] considers that BHT and BHA belong to groups 3 and 2B respectively, which correspond to “not classifiable as to its carcinogenicity to humans” and “possibly carcinogenic to humans”, respectively. Additionally, some research has evaluated the toxicity of synthetic antioxidants, considering the concentration of compounds that can cause the death of 50% of the test population (rats), called limit-dose 50 (LD_50_), represented by mg/kg body weight. Therefore, according to standard procedures in toxicology, the LD_50_ values for BHT, BHA, and TBHQ, shown in Table 1, indicate that these antioxidants can be considered moderately hazardous [8,40,41].

Recent reports have shown that BHA, if present in high concentrations, can affect the production and reactions with estrogen, resulting in a great impact on reproductive development, presenting carcinogenic and mutagenic effects [19,32], and is still to have a strong cytotoxic effect on humans [42]. Previous research have shown the toxicity of BHT in the development of zebrafish embryos/larvae [33,43]. Other studies have shown that BHT and these transformation products present an endocrine-disrupting effect, which promotes the development of tumors and cancer in humans [9,32,33] and may also have a potential teratogenic effect on aquatic organisms [44].

Research has also indicated that other synthetic phenolic antioxidants present toxicity. PG has testicular toxicity, inducing male infertility through mitochondrial dysfunction and disruption of calcium homeostasis [45]. TBHQ, an important antioxidant and also the main metabolite of BHA, has been proved to have an estrogenic effect, influencing the sexual formation of marine organisms [33].

Despite the numerous controversies about the toxicity of antioxidants, it is common sense that the use of these substances, mainly BHT and BHA, can represent dangers both to the environment and to humans, mainly due to their genotoxic effects and toxicity to biological organisms when found in concentrations beyond those recommended. Therefore, contaminations by antioxidants can result in public health problems, resulting in the need for extremely rigorous quality control of foodstuffs, which is only possible with analytical techniques capable of identifying and quantifying the presence of these synthetic antioxidants in complex matrices.

### 2.3. Detection of Antioxidants

Most countries have a firm commitment to promote the Sustainable Development Goals (SDG) proposed in 2015 by the United Nations, in which zero hunger and good health and well-being of people are SDG 2 and SDG 3, respectively; therefore, ensuring food safety and adequate nutritional values are key to achieving these goals. Therefore, it is essential that the regulatory agencies of food industries monitor the quality of foodstuffs, considering also the traceability from raw materials until the final foodstuffs, considering mainly the presence of contaminants and the adequacy of quality parameters, including antioxidant contents.

The evaluation of the appropriate type and level of antioxidants in foodstuff production, considering the manufacturing, processing, preparation, treatment, packaging, and transporting steps, and the general control of quality in foodstuffs requires adequate choices of analytical tools for the chemical analysis of antioxidants. For this, some preliminary steps should be considered, which include the sampling of the foodstuffs, the complexity of the extraction procedure to remove synthetic antioxidants from the matrix, the efficiency in the removal of the interference from the samples, and finally, the chemical analysis for the identification and quantification of the antioxidant [46]. This chemical analysis can be performed using separation and detection chromatographic or voltammetric techniques, which are based on the mobility of antioxidants between a stationary and mobile phase, followed by the detection in a specific detector or current measured after the application of potential difference.

Chromatographic or voltammetric techniques present different analytical parameters (precision, accuracy, linear range concentration, sensitivity, selectivity, and robustness), which must be taken into consideration before the choices of tool for antioxidant analysis. Besides, the number of synthetic phenolic antioxidants, the complexity of the foodstuff’s composition, and the necessity to employ analytical methods combined with advanced extraction techniques must be considered as typical factors in the evaluation of the efficiency of the antioxidant detection. An analytical method for simultaneous detection is also fundamental due to the industrial process often employing two or more types of antioxidants that allow desirable synergistic effects. Additionally, the available instrumentation and financial resources, as well as skilled labor, quick analysis, simplicity, reliability, and minimization, in toxic waste generation are important factors that must be taken into consideration before the analysis of foodstuffs in the control of quality or the contamination or adulteration detection by inadequate use of antioxidants.

A systematic search in ScienceDirect^®^ databases using the keywords “*analytical determination of synthetic antioxidants in foods*”, “*antioxidant*” plus “*chromatography*”, and “*antioxidant*” plus “*voltammetry*”, considering all published works between the years 2011 and 2022, was effectuated. Also were considered in this search the terms “*analytical determination*” plus the name of synthetic antioxidants and the combination of terms used early. All searches considered only research papers and the analytical procedures applied in foodstuff samples. In the center of Figure 2, it shows the perceptual values of published works employing chromatographic detection and voltammetric techniques, which correspond to around 40 papers.

The employ of chromatographic techniques, liquid or gaseous, requires previous pre-treatment and cleanup steps performed by different extraction procedures, which can result in various errors mainly by the manipulation of samples, high costs, and low efficiency of extraction, which can end up making the methodology insensitive and low selective. Therefore, many extraction procedures have been reported for synthetic antioxidant analysis, as demonstrated in Figure 1. Already, the success in the use of voltammetric techniques depends on the adequate choices of material for working electrode preparation, since the redox reaction occurs in the interface electrode/solution, and the change in the surface alters the sensitivity and selectivity. Thus, a great variety of materials have been proposed in the working electrode preparation for the voltammetric determination of antioxidants, as shown in Figure 2. Therefore, the application of chromatographic and voltammetric techniques, considering the main extraction procedures and materials for working electrode preparations, respectively, will be explained afterward.

## 3. Chemical Analysis of Synthetic Antioxidants in Foodstuffs

The information obtained by the systematic search in ScienceDirect^®^ databases, shown in Figure 2, indicated the predominance of the use of separation chromatography, followed by detection techniques in the development of analytical methods for the detection of phenolic antioxidants in foodstuffs. This is because these techniques allow the simultaneous determination of different types of antioxidants, through the separation of these compounds at different stages of a column, thus obtaining different retention times, which are related to the physicochemical characteristics of the antioxidants and their interaction between the stationary and mobile phase. After the separation, the antioxidants are identified and quantified using specific chromatographic detectors, such as ultraviolet–visible, diode array, thermal conductivity, and mass spectroscopy, resulting in a suitable sensitivity and selectivity [47].

However, separation and detection chromatographic, despite being very accurate in the detection of antioxidants, have as their main disadvantage the use of large amounts of organic solvents or inert gases, with elevated purity and, consequently, high cost. Additionally, the use of these techniques requires rigorous steps of extraction and cleanup to prepare the foodstuff samples for analysis, remove interference compounds, and/or preconcentrate the antioxidants to obtain reliable information. Extraction steps can increase the time and costs in the analysis, promote a reduction in the analytical frequency, and generate a great quantity of residues, which goes against a very important principle, taken very seriously today, green chemistry, which orients the reduction or elimination of toxic residues in chemical products and processes, including all cycles of a chemical, in its design, manufacture, use, and final disposal [48].

Voltammetric techniques, based on electrical potential measurements, are divided according to the mode of potential applications and, for the analysis of antioxidants, have been reported for the employ of differential pulse voltammetry (DPV), linear scan voltammetry (LSV), and square wave voltammetry (SWV), as shown in Figure 2. However, a potential constant application technique, such as multiple pulse amperometry (MPA), also has been reported. The sensitivity reported using DPV and SWV is similar to that obtained using chromatographic detection but presents as the main advantage the simplicity in the detection of phenolic antioxidants in foodstuffs due to the fact that complex and expensive extraction steps are not necessary, often requiring only one step of liquid–liquid extraction, possibly the synthetic antioxidant detection in the extract obtained from the initial sample, without the need for complex cleanup steps [49,50,51,52].

The analytical signals from the voltammetric determination of synthetic antioxidants are related to the redox reaction that occurs in the interface electrode/solution, and the reactants and/or products from the reaction can be adsorbed in the working electrode surface, resulting in a reduction in the intensity of analytical signals and difficulties in their reproducibility. Therefore, the adequate choice of material for the preparation of the working electrode is the limiting factor in the success of antioxidant detection. For this, the use of various types of working electrodes has been reported, including chemically modified surfaces by nanoparticles, polymers, and carbon forms, as shown in Figure 2. The following will report some characteristics of the chromatographic detection and voltammetric techniques employed in the determination of synthetic antioxidants in foodstuffs, indicating the applicability of extraction steps and adequate working electrodes.

### 3.1. Chromatographic Analysis of Synthetic Phenolic Antioxidants

The official methodology adopted by regulatory agencies around the world to identify and quantify the synthetic antioxidants in foodstuffs involves the use of previous gas chromatography (GC) and high-performance liquid chromatography (HPLC) for separations, followed by detection using different detector types [34,53,54,55].

HPLC can be applied in the separation of any compound that is soluble in a liquid phase, which can be modified according to the polarity of the blend antioxidant. The separation depends on the interaction of antioxidants with the mobile and stationary phase, which results in the retention times being similar, requiring specific extraction steps for improvement in the selectivity of the analysis. The detectors are generally nondestructive, and therefore, synthetic phenolic antioxidants can be collected for further analysis, which is very important in the traceability and adulteration evaluation; besides, the detection can be carried out at room temperature, avoiding the destruction of thermally sensitive antioxidants. However, the most common detector, the ultraviolet detector (HPLC/UV–VIS), presents low selectivity in synthetic phenolic antioxidant detection, and for this, in the last decades, the selectivity and sensitivity in the HPLC detection have been improved with the detection of mass spectrometry, which is used to determine the mass-to-charge ratio of ions from the ionization of molecules of interest. However, the HPLC analysis requires the use of a large number of organic solvents, and the analysis is usually very time-consuming, resulting in band broadening and, therefore, lower resolution in comparison with GC [48].

Bibai Du et al. [56] used high-performance liquid chromatography–tandem mass spectrometry (LC–MS/MS) in the determination of 20 synthetic antioxidants, including BHT. The separation used a gradient elution with a mobile phase constituted by ammonium acetate solution and methanol, and the stationary phase was the combination of two C_18_ columns, where one was used for chromatographic separation and the other was placed between the injector and the eluent mixer to eliminate possible background contamination of the chromatographic system. The detection was performed by mass spectrometry with ionization electrospray (ESI) and atmospheric pressure chemical ionization (APCI) for multiple reaction monitoring. Among the 20 synthetic antioxidants evaluated, 10 were detected in vegetable oil samples, 13 in powdered milk samples, and 9 in infant fruit puree samples. Considering BHT, the detection limit observed was 0.09 ng/mL, and recovery in vegetable oils, powdered milk, and baby fruit puree presented values between 70% and 80%, indicating that more effective extraction procedures are needed in these antioxidants analyses.

GC results in high sensitivity, excellent resolution, and good separation capacity. However, its use is only possible in the detection of volatile and thermally stable compounds. The mobile phase is an inert gas with a constant flow, and the stationary phase is a column thermostat, which can have variation in the temperature, resulting in suitable resolution and lower retention time. Detectors in GC are destructive, making their use unviable in the adulteration analysis; however, they can be useful for various foodstuff quality control laboratories.

Farajzadeh et al. [57] used GC coupled to a flame ionization detector (GC-FID) in the quantification of BHT and BHA in honey samples. For this, helium gas and a capillary column of poly dimethyl siloxane were used as mobile and stationary phases, respectively. With constant elution, the partition of antioxidants resulted in detection limits of 1.7 ng/mL for BHT and 4.1 ng/mL for BHA. These results were only achieved by the employ of a previous dispersive liquid–liquid extraction, allowing enrichment factors and extraction recoveries in the range of 144–186% and 72–93%, respectively.

Despite the high cost, GC coupled to a mass spectrometer (GC-MS) is a robust separation and detection technique that offers a superior signal-to-noise ratio, is easily automatized, and features fast data analysis, which provides comparatively more accurate and reproducible results. Therefore, Gupta et al. [18] determined phenolic antioxidants in packed fruit juices using gas chromatography–tandem mass spectrometry (GC–MS/MS), resulting in a sensitive, selective, accurate, and precise analytical procedure. However, these analyses were only possible after the use of the QuEChERS (quick, easy, cheap, effective, rugged, and safe) procedure of the extraction, which extracted simultaneously various phenolic antioxidants, removed matrix interferences, and enriched the antioxidants from the matrix in a single step.

The adequate choice of pretreatment, extraction, cleanup, and preconcentration steps will directly influence the instrumental performance, with either GC or HPLC detection. The extraction step employs specific solvents and adopts standard procedures, which can, sometimes, result in a high consumption of time, elevated cost, high level of contamination, and low extraction efficiency. Furthermore, these factors are intensely affected by the type and concentration of the extraction solvent, extraction temperature, extraction time, and extraction pH, as will be explained below [48,58,59].

### 3.2. Extraction Methods for Synthetic Phenolic Antioxidants

The quantification of synthetic antioxidants in foodstuffs is a problem in food quality control laboratories and regulatory, environmental, and health agencies due to the complexity associated with these matrices, especially those rich in lipids since they contain a mixture of synthetic phenolic antioxidants. The presence of vitamins, minerals, moisture, chlorophyll, other antioxidants, phenolic compounds, fats, and proteins, considered interferences, can directly influence the repeatability, reproducibility, sensitivity, and selectivity of the chemical analysis. Thus, phenolic antioxidant detection requires previous preparation to eliminate the possible interferences, extraction to isolate the antioxidant in a suitable medium, and cleanup steps before chromatographic determination. In some samples, are necessary the concentration of antioxidants due to their presence at low concentrations, low levels of detection required by regulatory agencies, and the complex nature of the matrices that are present.

However, the extraction step, besides isolating the synthetic antioxidant of interest, can also remove other components of the sample, following damage in the chemical analysis. Therefore, in recent years, great advances have been noted in the development of suitable extraction methods that provide high recovery and reproducibility, in addition to being faster, cheaper, more ecological, and easier to automate [58]. In Table 2 are shown some extraction methods employed in the analysis of the synthetic phenolic antioxidants, indicating the extractant, the type of sample, the detection technique, the main experimental conditions, the limit of detection, and recovered values.

All reports indicated that minimizing the preparation steps mainly reduces sources of errors and time and cost in the chemical analysis. Additionally, it was observed that the major trends in sample preparation focused on miniaturization, automation, high-throughput performance, online coupling with analytical instruments, and low-cost operations with extremely low or no solvent consumption [1,25]. The following are some of the extraction methods more frequently employed in the analysis of the synthetic antioxidant in foodstuffs.

#### 3.2.1. Liquid–Liquid Extraction (LLE)

Liquid–liquid extraction (LLE) is an extraction method widely used in the past in complex sample extraction, separation, and purification processes, mainly due to its simplicity, efficiency, and ease of application in standard analytical methods. It is a technique that separates the analyte of interest from a complex matrix, based on the difference in solubility of the analyte, in two solvents of different polarities [60,61,62,63].

Kim et al. [64] evaluated a method for the determination and validation of an uncertainty measurement for the simultaneous determination of the synthetic phenolic antioxidants BHA, BHT, OG, PG, and TBHQ in edible oils commonly consumed in Korea. Antioxidants were extracted from the samples using 20 portions of hexane-saturated acetonitrile, in which the acetonitrile phase was collected and evaporated, and the extracts obtained were analyzed using LC–MS/MS under previously optimized conditions. Antioxidant recoveries ranged from 91.4% to 115.9% with relative standard deviations between 0.3% and 11.4% and uncertainties from 0.15% to 5.91%, indicating that the method is suitable for verifying the safety of edible oil products containing residues of these antioxidants.

However, as shown in the research above and in other published works, the efficiency of LLE is directly related to the use of large amounts of toxic and flammable organic solvents. Furthermore, due to the low concentration of the antioxidants in the obtained extract, the preconcentration step is necessary in the extract obtained. The use of LLE has been much questioned in recent years due to the need for large amounts of solvents, and the extraction is laborious, poses a risk to user safety, presents difficulty in automatization, and in some cases, occurs the formation of emulsions, damaging the separation of the organic from the aqueous fractions and, consequently, making difficult the extraction and detection of the antioxidant.

**Table 2 molecules-27-07137-t002:** Information about the synthetic phenolic antioxidant detection indicating the extraction method, extractant, type of sample, technique, experimental conditions, limits of detection, and recovery values. These pieces of information were obtained from scientific databases in the years 2011 to 2022.

Extraction Method	Extractant	Sample	Antioxidant	Technique	Experimental Conditions	LOD	LR	Rec. (%)	Ref.
LLME	Organic solvent	Soya bean oil Peanut oil Cereal cooking oil	BHT BHA TBHQ	GC/MS	DB-17 capillary column Helium carrier gas	0.001 mg/L 0.002 mg/L 0.004 mg/L	0.01–20 mg/L	94–108	[16]
QuEChERS	Adjust of pH NaOH MgSO4, NaCl Acetonitrile PSA	Fruit juice	BHA BHT TBHQ PG OG	GC/MS/MS	TG-5MS capillary column Helium carrier gas	8.14–25.45 μmol/L	100–1000 ng/L	80–115	[18]
DLLME	NR	Corn Sunflower Olive Canola Grape seed	BHT BHA TBHQ	GC/FID	HP-1 capillary column Helium carrier gas	0.32 ng/mL 0.42 ng/mL 0.13 ng/mL	0.13–0.42 ng/mL	NR	[7]
SBSE	PEDOT/MIL-101/PVA	Fruit juice Milk Infant formula Coffee Creamer	BHT BHA TBHQ	HPLC/UV	C18 column Water and acetonitrile	0.05–0.15 μg/kg	0.2–200 μg/kg	87–101	[65]
LDS-DLLME-MSPE	Organic solvent, DA@Fe_3_O_4_	Edible oil	BHT BHA TBHQ PG	HPLC/DAD	Reversed-phase C18 Column Methanol and water with 0.5% acetic acid	1.2–5.8 ng/mL	50–2000 ng/mL	90–100	[19]
WSVAME	Surfactant	Edible oil	BHA TBHQ	HPLC/UV		20–26 μg/L	0.2–200 μg/mL	95–102	[14]
SPE	Graphene	Precooked food	BHA PG	HPLC/UV	Chromaphase RP-18	12 mg/L 14 mg/L	0.4–16 mg/L	87–89	[13]
VACPE	Nonionic surfactant	Spices	BHT BHA TBHQ PG	HPLC/DAD	Acetonitrile and distilled-deionized water contained 0.1% acetic acid	3.2–9.8 ng/mL	8.0–800 ng/mL	89–103.5	[66]
LLE	Iso-propanol/n-hexane	Blend oil Olive oil Prickly oil Ginger oil Garlic oil	PG TBHQ BHA OG BHT	HPLC/FLD	WondaSil C18 column Methanol, acetonitrile, and 1% formic acid in water	NR	0.28–31.36 μg/L	97–108	[67]
UALLME	Organic solvent	Edible oil	BHT BHA TBHQ	GC/MS	HP-5MS column Helium carrier gas	0.04 ng/mL 0.03 ng/mL 0.04 ng/mL	1–50 ng/mL	86–115	[68]
LLE	Organic solvent	Edible oil	PG TBHQ BHA OG DG BHT	HPL/-UV	C18 column 5% acetic acid in Acetonitrile, 5% acetic acid in water	0.11–0.98 μg/mL	0.78–100 lg/mL	91–116	[64]
LLE-DSPE	Organic Solvent PSA GCB	Edible oil	BHT BHA	GC/MS	Helium carrier gas	0.002–0.04 mg/kg	0.5–20 mg/kg	74–118	[17]
DLLME	Organic Solvent Centrifugation	Fruit juice	BHT BHA	HPLC/UV	C18 reversed-phase column Methanol and water glacial and acetic acid	0.9 μg/L 2.5 μg/L	2–2500 μg/L	95–100	[69]
QuEChERS	NR	Salmon silage	BHA PG	HPLC/MS/MS	Ascentis^®^ Express C18 column Methanol/95%, water, and 5 mM ammonium formate in methanol	0.12–0.15 mg/kg	0.01–10 µg/mL	97–101	[70]
SPME	PTFE-faced septum	Beverages	BHT BHA TBHQ	GC/MS/MS	Rtx-1301 capillary column Helium carrier gas	0.005 μg/L 0.025 μg/L 0.05 μg/L	0.005–0.2 μg/L	98–109	[71]
CPE	Tergitol TMN-6 (TMN-6) nonionic surfactant	Edible oils	BHT BHA TBHQ PG	HPLC/UV	Reversed-phase C18 methanol and water with 1.5% acetic acid	1.6–9.0 ng/mL	1.0–500 ng/mL	90–98	[72]
DLLME	Organic Solvent Centrifugation	Honey	BHT BHA	GC/FID	DB-1 capillary column Helium carrier gas	1.7–41 ng/mL	5.0–20,000 ng/mL	144–186	[57]
SBSE-TD	NR	Soft drink	BHT BHA TBHQ	GC/MS/MS	HP-5MS column Helium carrier gas	0.03–0.05 ng/mL	0.5–20 ng/mL	81–117	[15]
UALLME	Organic Solvent Centrifugation	Edible oils	TBHQ	HPLC/UV	C18 reversed-phase column Methanol and 0.5% acetic acid aqueous solution	0.02 μg/mL	5–500 μg/mL	99–112	[20]
LLE	Organic Solvent Ultrasonic	Sunflower oil Olive oil	BHA PG TBHQ	Microchip capillary electrophoresis	Borate buffer pH 8.5	0.8–4.3 μmol/L	10–200 μmol/L	94–106	[73]
LLE	Organic Solvent Centrifugation	Sesame oil	BHA TBHQ	HPLC/CL	Methanol and water (80:20 *v*/*v*)	0.024 μg/mL 0.033 μg/mL	0.1–10 μg/mL	98–102	[74]
LLE	Organic Solvent Centrifugation	Edible oil	BHT BHA	SWV	AuNPs/GCE BR buffer pH 2.0	0.039–0.08 μmol/L	0.10–1.50 μmol/L	96–101	[75]
LLE	Organic Solvent Ultrasonic Centrifugation	Olive oil Peanut oil Potato chips Cookies	BHA PG	LSV	Nafion/SAP/ HRP/Au- GN/GCE BR buffer pH 2.0	0.046 mg/L 0.024 mg/L	0.1–100 mg/L	87–126	[76]
LLE	Organic solvents	Edible oil	BHA TBHQ	LSV	AuNPs/ERGO/GCE PBS, pH 7.0	0.23 μmol/L 0.31 μmol/L	1.0–10 μmol/L	NR	[77]
LLE	Organic Solvent Centrifugation	Chewing gum	TBHQ BHA PG	MPA/FIA	Carbon glassy BR buffer, pH 2.0	NR	NR	95–116	[78]
LLE	Organic Solvent Centrifugation	Vegetable oil	TBHQ BHA	SWV	ZnO TPHS@GO/GCE citric acid, Na_2_HPO_4_ buffer, pH 3.0	0.14 μmol/L 0.05 μmol/L	0.30–65 μmol/L	95–106	[49]
LLE	Organic Solvent Centrifugation	Potato chips	BHA	DPV	NiHCF modified GWCE PBS, pH 7.0	0.6 μmol/L	1.2–107 μmol/L	98–100	[50]
LLE	Bz + EtOH binary mixture (1:2) in 0.1 mol/L H_2_SO_4_	Edible oil	BHT BHA TBHQ	SWV	Ultramicroelectrode PBS, pH 7.0	NR	NR	89–118	[51]
LLE	Organic Solvent Centrifugation	Ghee Sunflower oil Salad dressing	BHT BHA TBHQ	LSV	AuNPs/graphite	0.4 μmol/L 0.1 μmol/L 0.6 μmol/L	NR	90–106	[79]
LLE	Organic Solvent Centrifugation	Potato chips	BHA	DPV	POC/MWCNT	0.11 μmol/L	0.33–110 μmol/L	98–105	[52]
LLE	Brij^®^ 35	Linseed oils	BHA TBHQ	DPV	MWNT-Brij^®^ 35 modified glassy carbon electrode in Brij^®^ 35 micellar medium LiClO_4_ with Brij^®^ 35	0.26 μmol/L 0.15 μmol/L	1.0–1000 μmol/L	99–103	[80]
LLE	Organic Solvent	Dry potato flakes	BHA	AMP	CuHCFNP/EMIMBF_4_ gel-modified electrode Phosphate buffer, pH 7	0.5 μmol/L	1.5–1000 μmol/L	97–99	[81]
LLE	Organic Solvent	Linseed oils	BHA TBHQ	DPV	poly-carminic acid/MWNT/GCE Britton–Robinson buffer (BR) pH 2.0	0.23 μmol/L 0.36 μmol/L	1.5–100 μmol/L	100–103	[82]

Legend: NR: not reported; LOD: limit of detection; Rec.: recovery; Ref.: references; injector port silylation–gas chromatography–tandem mass spectrometry (GC/MS/MS), dumbbell-shaped stir bar adsorbent of MIL-101 entrapped in PVA cryogel coated with poly(3,4-ethylenedioxythiophene) (PEDOT/MIL-101/PVA), stir bar sorptive extraction (SBSE), water-contained surfactant-based vortex-assisted microextraction (WSVAME), low-density solvent-based dispersive liquid–liquid microextraction coupled with magnetic solid-phase extraction (LDS-DLLME-MSPE), decanoic modified magnetic nanoparticles (DA@Fe_3_O_4_), dispersive liquid–liquid microextraction (DLLME), vortex-assisted cloud-point extraction (VACPE), high-performance liquid chromatography with a fluorescence detector (HPLC/FLD), primary secondary amine (PSA), ultrasonic-assisted liquid–liquid microextractions (UALLME), stir bar sorptive extraction and thermally desorbed (SBSE-TD), cloud-point extraction (CPE), glassy carbon electrode surface multiwall nanotubes poly-carminic acid (poly-carminic acid/MWNT/GCE), multiwalled carbon nanotube (MWNT), copper hexacyanoferrate nanoparticles (CuHCFNP) 1-ethyl-3-methylimidazolium tetrafluoroborate (EMIMBF_4_) (CuHCFNP/EMIMBF_4_), PC matrix (Plasticyl^®^ PC 1501)/multiwall carbon nanotube (PC/MWCNT), gold nanoparticles (AuNPs), spiny Au-Pt (SAP) horseradish peroxidase (HRP) gold nanoparticles modified graphene (Au-GN) glassy carbon electrode (SAP/HRP/Au-GN/GCE), LSV: linear sweep voltammetry, SWV: square wave voltammetry, MPA/FIA: multiple-pulse amperometry with flow injection analysis, AMP: amperometry, LR: linear range.

#### 3.2.2. Solid-Phase Extraction (SPE)

Solid-phase extraction (SPE) is an extraction method that emerged in the 1970s, used to isolate and concentrate the analyte of interest from a sample by retaining it in a solid phase, which will later be recovered by elution from a liquid or fluid or by thermal desorption in the gas phase. The main advantages of SPE over LLE are the simplification of the trace enrichment matrix (sample cleaning) and medium change (transfer of the sample matrix to a different solvent or gas phase) and use of a smaller volume of solvent, in addition to being easy to set up.

SPE is widely used in chemical analysis, including environmental, pharmaceutical, clinical, food, and industrial applications. Over time, various sorbent formats were developed to facilitate the extraction of different sample types. Furthermore, it is a technique that allows its automation, so a full system integration for autonomous extraction, separation, and detection is possible, although flexible and cost-effective manual sample processing remains the most common practice in food analysis [83,84].

Many types of sorbents are commercially available and used in food analysis, such as alumina, magnesium silicate, and graphite carbon; however, silica is the most used because its surface is more reactive, allowing modifications by chemical reactions to improve the extraction specificity in addition to being stable. However, when used in solid matrices, sample treatment steps, such as homogenization, filtration, sonication, centrifugation, and liquid–liquid pre-extraction, are sometimes used, which makes the technique extremely time-consuming and expensive. Another disadvantage of this technique is that the cartridge must be uniformly coated to avoid reproducibility difficulties in the analytical determination step [48,84].

In the last years, graphene has also been employed as a sorbent for SPE in foodstuff samples, as performed by Mateos et al. [13], who evaluated the usefulness of graphene as a sorbent for the isolation of BHA and PG from samples of precooked spaghetti and hard bouillon cubes. Therefore, SPE was influenced by the type and volume of the eluent, the amount and volume of graphene, and the concentration of antioxidants in the samples. HPLC analysis indicated that PG and BHA levels were below the established legal limits. Hard bouillon cubes, free from both antioxidants were fortified with different amounts of PG and BHA, and recoveries very close to 100% were achieved, demonstrating that graphene is a suitable sorbent in the SPE extraction of antioxidants in foodstuffs.

#### 3.2.3. Ultrasound Extraction

SPE and LLE have several associated disadvantages, such as high capital investment and energy consumption and the use of toxic organic substances used for extraction. For this, ultrasound-assisted extraction (UAE) is suitable in terms of being environmentally friendly and having clean extraction and low-investment-required technique. Additionally, UAE is easy to use, multidirectional, and flexible when compared with other extraction techniques. High-intensity sonication is performed for extraction and process applications, while low-intensity sonication is used as a nondestructive analytical technique for quality assurance and process control. Ultrasound application enlarges the solvent selection range of generally recognized safe, instead of toxic, organic solvents [48,85].

Cacho et al. [15] used UAE in the extraction of BHA, BHT, and TBHQ in edible vegetable oil, resulting in a recovery range of 86% to 115% in the extracts. Žnideršič, Mlakar, and Prosen [71] used UAE in the degasification of beer, before BHT, BHA, and TBHQ extraction, which was followed by GC–MS/MS analysis, resulting in limit quantification between 0.08 and 0.10 ng/g, depending on the compound.

#### 3.2.4. Solid-Phase Microextraction

In recent years, many innovations in analytical processes have also been applied in the field of sample preparation, which has resulted in the replacement of classic methods with faster, cheaper, less toxic procedures with less waste generation and with equal or better performance than classical methods, called the green extraction methods. Current trends in sample preparation have focused on low-cost operations, moving towards miniaturization, automation, high-efficiency performance, online analytical instruments, and extremely low or solvent-free consumption. Reducing the sample preparation steps can be effective in the errors, time, and cost decrease, and has some advantages for measuring trace and ultratrace analytes in complex matrices. For this, microextraction methods, such as solid-phase microextraction (SPME), stir bar boundary extraction (SBSE), and liquid-phase microextraction (LPME) have become important for sample preparation compared with conventional techniques. Microextraction means that all modes of these techniques require the use of small volumes of extraction medium during extraction conditions [48,86].

SPME is a sample preparation method that uses fused silica-coated externally with an appropriate stationary phase, which, during the elution of the sample, is retained in the coating and can later be extracted. The column used in this technique is infinitely smaller than that used in conventional SPE, in addition to allowing you to compress all the steps of sample preparation into one. This considerably reduces preparation time and solvent use and, in addition to cost, generates less waste for disposal. It can be easily coupled with GC and HPLC chromatography analysis and improve detection limits [62,87].

SPME has been successfully applied to a wide variety of compounds in gaseous, liquid, and solid samples, especially for the extraction of volatile organics and semivolatile compounds in environmental, biological, and foodstuffs samples, followed by GC and GC/MS analysis. Furthermore, SPME has also been introduced for direct coupling with HPLC and LC/MS to analyze weakly volatile or thermally unstable compounds and various polar compounds. These SPME methods are based on the adsorption of compounds in the liquid phase coated on the fiber surface [48,62,87].

Žnideršič et al. [71] employed SPME in the extraction of BHT, BHA, and TBHQ in beverages previously degassed. Antioxidants were adsorbed in an SPME fiber, the volatile compounds were removed under reduced pressure, and the extract desorbed was analyzed by GC–MS/MS, resulting in limits of detection of 0.005, 0.025, and 0.005 μg/L for BHT, BHA, and TBHQ, respectively, and recovery between 98% and 109% in beverages.

#### 3.2.5. Stir Bar Sorptive Extraction

The stir bar sorptive extraction (SBSE) is an extraction technique developed in the late 1990s based on SPME. The larger coating on the stir bar, about 50 to 250 times larger than the fiber used in SPME, allows for greater adsorption and recovery capacity, ensuring greater efficiency and better reproducibility in the extraction of the compounds of interest from complex samples. In addition, it significantly reduces the use of solvents and, due to its greater absorption power, reduces the amount of sample preparation, which is often time-consuming and laborious and generates experimental errors. Normally, SBSE can be applied in the extraction of various organic compounds in the aqueous matrix of foodstuffs. Therefore, the extraction bar can be added to the sample and by the stirring process and carry out the extraction process. After the extraction time, the compounds adsorbed on the bar can be desorbed and determined by GC or HPLC. The biggest disadvantage of this technique is that it is not possible to automate it [48,62].

Nurerk et al. [65] optimized an extraction methodology using SBSE to extract BHA, BHT, and TBHQ from juice, milk, infant formula, and coffee cream samples. For this, it employed a stir bar adsorbent trapped in poly(3,4-ethylenedioxythiophene) interconnected porous cryogen, and the extraction efficiency was optimized by evaluating the effect of adsorbent compositions, extraction time, agitation speed, sample pH, desorption conditions, sample volume, and ionic strength. The analysis of the extracted synthetic phenolic antioxidants was performed by HPLC, and the detection limits for BHA, BHT, and TBHQ ranged between 0.05 and 0.15 μg/kg, and recoveries ranged from 87% to 101%. The developed composite stir bar adsorbent was convenient to use with good physical and chemical stability, allowing efficient extraction for 12 cycles of extractions.

#### 3.2.6. Liquid–Liquid Microextraction

Liquid-phase microextraction (LPME) is an extraction technique that promotes the preconcentration and separation of the analyte from its matrix widely used in recent years, mainly due to its simple operation, low cost, and high efficiency. LPME is an easy, fast, efficient, and cost-effective sample preparation technique. Like SPME, sample extraction, concentration, and entry can be integrated into a single step. LPME extraction typically consists of a small amount of a water-immiscible solvent and an aqueous phase containing the target analyte. The acceptor phase is not only immersed for direct extraction but also suspended in the sample for headspace extraction, and the received phase volume varies in microliters or less; besides, greater enrichment factors must be obtained due to the relationship between the high volume of the sample and the acceptor phase [48,62,88].

LPME can be classified into single-phase droplet microextraction (SDME), hollow fiber liquid-phase microextraction (HF-LPME), and dispersive liquid microextraction (DLLME), according to the operation mode. Therefore, different LPME approaches have been developed to analyze various compounds in foodstuffs, with each group having a variety of modifications. The advantages of LPME can be summarized as a simple and highly selective extraction method, environmentally friendly due to lower solvent usage, where the μL of the solvent is used to extract an analyte from multiple samples and mainly can be combined with HPLC, GC, and capillary electrophoresis (CE) [48,62,88].

Liu et al. [20] used LPME allied to ultrasound-assisted and deep eutectic solvents for the extraction of TBHQ in edible oils, in which the extracts were directly analyzed HPLC with an ultraviolet detector after the separation by a reverse-phase column. In this methodology, they obtained a detection limit of 0.02 μg/mL and a recovery range of 98.5%–112%. This methodology was applied to determine TBHQ in 13 edible oil samples, and the result was close to that of the conventional LLE, but with the use of toxic solvents, which proves to be an environment-friendly method.

DLLME, first described in 2006, has been successfully applied to extract pesticides from water samples. Like LLME, DLLME relies on three-component solvent systems (aqueous sample, dispersive solvent, and extractive solvent). A suitable mixture of extraction solvent (organic) and dispersive solvent (water-miscible organic solvent) can be injected into the aqueous sample, and therefore, turbid solvent must be formed. Subsequently, using a centrifuge, the analytes are separated from the organic phase. DLLME’s main advantages over conventional techniques are simplicity, fast operation, low cost, easy handling, low use of organic solvents, high recovery, high factor enrichment, and adaptability to HPLC and GC techniques [48,89,90].

DLLME was used by Biparva et al. [69] in the extraction of BHA and BHT in fruit juice samples for HPLC analysis. The detection limits obtained were 2.5 and 0.9 mg/L for BHA and BHT, respectively, with recovery percentages between 95% and 100% after the preconcentration of BHT and BHA. Already, Fang et al. [17] used DLLME as a cleanup step in the edible oil sample analysis, and HPLC detection resulted in detection limits from 0.002 to 0.04 mg/kg and a recovery percentage of around 74% in the oil samples. This method resulted in less time and less consumption of reagents that could be used on several occasions to detect synthetic antioxidants.

#### 3.2.7. Cloud-Point Extraction

Cloud-point extraction (CPE) is the more ecological sample pretreatment, which consists of three steps: (i) solubilization of the analytes in micellar aggregates, (ii) cloudiness, and (iii) phase separation for the analysis. Nonionic surfactants may be able to form a micelle in aqueous solutions and become cloudy at a specific temperature, which is described as cloud-point temperature. At this point, the micellar solution is divided into two phases: a small-volume phase that enriches in terms of surfactant and dilutes the aqueous phase. When metal ions react with an appropriate binder, they can form an aqueous complex of low solubility, and therefore, these ions must be extracted from the aqueous solution in the small-volume phase enriched in terms of surfactant [91,92,93].

CPE is a simple, sensitive, and fast method of concentration and separation of essential elements because of employing water and avoiding the use of expensive, toxic, and flammable organic solvents in large volumes. In addition, CPE is expected to have several significant advantages, such as faster operation, easier handling, shorter time, lower cost, higher recovery and enrichment factor, and less stringent requirements for separation. Diluted surfactant solvents can be used as an extracting medium in CPE, resulting in less laboratory waste and cost-effectiveness, and are likely to be cost-effective reagents. Furthermore, surfactants are less flammable than organic solvents, reducing the risk to the analyst [91,92,93].

CPE was employed by Chen et al. [69] to extract and preconcentrate PG, TBHQ, BHA, and BHT from edible oils. Using a nonionic surfactant and assisted ultrasound, these antioxidants were extracted and analyzed by HPLC after the separation in a reverse-phase column, following limits of detection in the range of 1.6 to 9.0 ng/mL and a recovery range of 90% to 98%. The comparison with the CPE method using neutral surfactant and LLE using methanol proved that the proposed method allows a preconcentration factor of 25 times, improving the analytical sensitivity.

#### 3.2.8. QuEChERS

A fast, easy, cheap, effective, robust, and secure (QuEChERS) approach can extract multiclass analytes simultaneously, remove matrix interferences, and enrich matrix analytes in a single step. This approach has advantages; for example, (i) it reduces analysis time and cost, (ii) it requires a small number of steps, and (iii) it minimizes the consumption of chemicals. Initially, the QuEChERS method was used in different matrices for analysis in food, biological, and waste industries. QuEChERS extraction is divided into two stages, an initial single-phase extraction with a solvent, followed by salting-out extraction/partitioning with salts, and finally, a dispersive solid-phase extraction to clean the extract (due to possible interferences present as a consequence of complex matrices) [18,94].

Gupta et al. [18] employed QuEChERS as an extraction procedure of synthetic antioxidants in fruit juice samples for GC–MS/MS analysis. With a detection limit of 8.14 to 25.45 ng/L, the proposed method was successfully applied to six different packaged fruit juice samples, resulting in a recovery in the range of 73.2% to 119.9%, thus being considered effective and cost-effective for routine antioxidant detection in foodstuff samples.

In another research, Guldberg et al. [70] employed QuEChERS for the extraction of BHA and PG in fish silage and fish oil used to produce animal feed. This extraction was performed to develop and validate a new method to determine these antioxidants by LC–MS/MS and allowed for the quantification of antioxidants in all matrices with low detection limits of 0.012 to 0.015 mg/kg and a recovery range of 97% to 101%.

### 3.3. Electrochemical Analysis of Synthetic Phenolic Antioxidants

Electrochemical analyses are based on electrical property measurements, such as current, potential, conductivity, and impedance among others, which allow us to quantify the compounds of interest. The mode of potential application in the function of time is typical for each electrochemical technique, resulting in different signals’ profile and, consequently, different analytical parameters, in which the voltammetry presents the best sensitivity and applicability.

Voltammetric techniques are based on measurements of electrical currents that flow through the system because of the application of an electrical potential difference, which has enough energy to promote oxidation and/or reduction reactions (redox reactions) of the chemical species of interest. Thus, the current generated is directly proportional to the concentration of electroactive species present in the samples, and the electrical potential is related to the identity of the species of interest [95].

The use of voltammetric techniques results in fast analysis, low-cost instrumentation, and maintenance, and mainly, easy instrument handling and data processing. Considering the quality control of products and processes, they are techniques capable of identifying and quantifying low concentrations of organic and inorganic compounds. Voltammetric techniques are most promising in chemical analysis in general and can replace conventional techniques (chromatographic separation and detection and spectroscopic techniques) in industrial processes and quality control in chemical, pharmaceutical, and food industries, as well as in environmental analysis [96].

Voltammetric techniques can be robust and allow low cost and fast analysis since they require simple and fast sample extraction steps, even in the case of very complex samples, such as foodstuffs, in addition to samples containing suspended solids, which are compounds that interfere with sensitivity and selectivity in other methods, as is the case with ascorbic acid in juice, which can mask the existence of other organic acids at lower concentrations, or when working with colored samples that do not allow direct measurement by spectrophotometric and chromatographic techniques, as in the case of wines and juices of fruits with a high content of anthocyanins [97].

Consequently, in the last two decades, voltammetric techniques have been reported as a suitable alternative in foodstuff analysis in either the identification of the chemical composition, quality control for food safety, traceability, and adulteration in raw materials and industrialized products. Furthermore, the voltammetric techniques can be used with success in the determination of the synthetic phenolic antioxidant due to these compounds having chemical structures containing electroactive groups that can be electrochemically reduced and/or oxidized, resulting in currents and potential values that can be used in the quantification and identification of an antioxidant. The redox reaction occurs in the interface between the solution containing the antioxidant and the working electrode surface, and resulting voltammograms present a profile related to how the electrical potential difference was applied and intensities’ currents related to the concentration of the compound of interest.

Additionally, the electrochemical mechanism type of electron transfers (chemical, electrochemical, and coupled chemical reactions) and the experimental (solvent, electrolyte, and pH) and voltammetric (interval potential, scan rate of potential, pulse intensities, and directions, among others) optimized parameters will influence intensely the sensitivity in voltammetric analysis. Furthermore, the use of voltammetric techniques allows the acquisition of many experimental parameters that help in the physicochemical characterization, such as antioxidant activity, chemical equilibrium constant, chemical stability, and kinetics of the reaction, of the phenolic compounds, natural or synthetic, present in foodstuff samples. Some of these pieces of information are relevant for designating which species have the greatest contribution to antioxidant activity and, consequently, their efficiency in foodstuff production [98].

The use of modern instrumentation allows analytical results with a sensitivity comparable to those obtained by chromatographic techniques, mainly with pulse voltammetric techniques, such as differential pulse (DPV) and square wave voltammetry (SWV). Their use does not require complex extraction and cleanup steps, and for this, these voltammetric techniques have been largely employed in synthetic antioxidant determination considering foodstuff sample, as shown in Table 2 and described below.

In DPV, a series of potential pulses with a constant amplitude is applied, superimposed on the linear ramp. The currents are measured two times, at the beginning and the end of each pulse, and the voltammogram is recorded from the difference between these two current values as a function of the applied potential [99]. Therefore, Farajmand et al. [100] used DPV combined with the previous LPME for the determination of TBHQ in edible oil samples, resulting in a dynamic range of 5 to 200 mg/kg and a limit of detection of 1.8 mg/kg, so this method was agreeable and comparable to those achieved using HPLC/UV–VIS.

SWV presents the best sensitivity among all voltammetric techniques and has been reported in the analysis of synthetic antioxidants in foodstuffs. In SWV, a series of cathodic and anodic pulses with equal amplitude are applied under a staircase potential, which promotes reduction and/or oxidation reaction, and the resultant signal of current is the difference between both currents measured at the final of each pulse, resulting in a significant increase in the sensitivity [96,99].

Robledo et al. [101] used SWV allied to a carbon fiber disk ultramicroelectrode in the detection of BHT extracted with acetonitrile from edible vegetable oils, resulting in concentrations calculated in good agreement with those values declared by the manufacturers.

One of the main challenges for the determination of synthetic phenolic antioxidants in foodstuffs is that a mixture of two or more antioxidants is usually used in the analyzed samples, which, due to their chemically similar structures, make it difficult to analyze low concentrations without using the previous steps of separation. However, voltammetric techniques make it possible to study the redox profile of each of the antioxidants, the redox processes, the combined flow injection techniques, and the use of carbon electrodes from different sources, such as glassy carbon and boron-doped diamond [102], and chemically modified electrodes [50,79].

Voltammetric studies of the BHA, TBHQ, and PG oxidation process were carried out to determine this antioxidant in chewing gum [78,103]. BHA and BHT were determined simultaneously in enriched samples of dehydrated potato flakes using SWV and DPV combined with the use of cylindrical carbon fiber microelectrodes, resulting in a difference in reaction redox of about 300 mV, considering the peak potential values [104]. The use of a carbon paste electrode modified with nickel phthalocyanine was also reported for the catalytic voltammetric study and simultaneous determination of BHA and BHT [105].

Some reports have indicated that the use of the previous preconcentration step before the DPV or SWV measurement can enhance the sensitivity due to the previous accumulation of synthetic phenolic antioxidants under the working electrode. Therefore, Alipour et al. [104] used DPV and performed a previous preconcentration of GA under an activated pencil lead electrode and observed that the limit of detection of 0.25 µmol/L was improved to 5.2 nmol/L, allowing the voltammetric determination of gallic acid in black and green tea and mango juice samples.

Linear sweep voltammetry (LSV), where the current is measured when the electrode potential is varied linearly with time, begins in a value where no reaction occurs and follows until a value where reduction or oxidation reactions occur has been reported in some research for antioxidant detection [96,99]. However, LSV results in low sensitivity when compared with pulse techniques, such as DPV and SWV.

Furthermore, the use of other electroanalytical techniques that involve the application of a constant potential value has also been reported, such as chronoamperometry (CA). It is related to the measurement of the current after the application of fixed potential promotes redox reaction and is frequently used as a detection technique in chromatography and capillary electrophoresis, in electroanalysis, it is used when enzyme-based biosensors, immunosensors, and aptasensors are working electrodes.

### 3.4. Working Electrodes Used in Synthetic Phenolic Antioxidant Analysis

Like other organic compound detections, for the electrochemical determination of synthetic phenolic antioxidants, the main step that will determine the success of the methodology is the surface on which the redox reaction will occur; already in the use of chromatographic techniques, the determining step of the methodology is the extraction, as shown in Figure 1 and discussed early. For electrochemical analysis for all the articles found in the last few years, just a simple liquid–liquid extraction was enough, without additional cleanup steps to determine the antioxidants in food.

In practice, for electrochemical analysis, the sample containing the compounds of interest is added in the electrochemical cell, which is composed of electrical conductors (working, reference, and auxiliary electrodes) and the ionic conductor (supporting electrolyte). The potential difference is applied, resulting in an interfacial reaction that occurs between the working electrode surface and a solution.

Therefore, the choices of working electrodes are based on the chemical structure of antioxidants, which will result in a specific potential interval where the antioxidant oxidizes and/or reduces, following the signal proportional to the antioxidant concentration. Additionally, the working electrodes must present low cost, high surface area, and good electric conductivity. Besides, the signal-to-noise features and reproducibility in the signals and still results in voltammetric methodologies with suitable robustness, efficiency, sensitivity, inexpensiveness, and aim at meeting green chemistry principles should also be considered [97].

The preparation of working electrodes requires a specific procedure according to the material employed, allowing for specific protocols that combine the surface cleaned by specific solvents or surfactants, alteration of the exposed microstructure by mechanical polishing, manipulation of the surface chemistry by an electrochemical process that can improve the intensity and reproducibility of the analytical signal [96]. Additionally, working electrodes with different types of materials and geometries are commercially available, and many can be employed for surface modification and specific uses.

Carbon is the most common material used as a working electrode due to its broad anodic potential range, low cost, chemical inertness, and mechanical stability. There are different forms of carbon, among them the most used are graphite, glassy carbon, carbon fiber, nanotubes, graphene, and diamond.

Medeiros et al. [106] developed a methodology for the simultaneous determination of BHA and BHT in foods using SWV combined with a boron-doped diamond (BDD) electrode. Through the evaluation of the supporting electrolyte and suitable voltammetric parameters, it was possible to observe that SWV together with a cathodically pretreated DDBE electrode can be used with some benefits for the quantitative determination of BHA and BHT, alone or mixed as commonly found in food products. Very low limits of detection were obtained in the simultaneous determination of BHA (0.14 μmol/L) and BHT (0.25 μmol/L). Furthermore, addition and recovery analysis allowed us to conclude that the matrix effect did not present significant interference. The concentration values obtained for BHA and BHT are like those obtained by the HPLC method. Thus, the SWV method reported here is effective for the simultaneous determination of BHA and BHT in food products.

Bavol et al. [78] presented the applicability of multiple-pulse amperometric detection combined with the GC electrode as a working electrode coupled with flow injection analysis (FIA) combined with multiple pulse amperometry for the simultaneous determination of TBHQ, PG, and BHA antioxidants. The technique provides a short analysis time and low consumption of reagents and samples, in addition to high precision. The limits of quantification were 2.51, 1.45, and 0.85 μmol/L for TBHQ, PG, and BHA, respectively. Furthermore, the method requires simpler instrumentation and lower investment and operating cost compared with other more expensive techniques, such as HPLC. The methodology developed was used in the determination of antioxidants in samples of chewing gum, requiring only a simple extraction process, demonstrating the robustness and efficiency of the methodology.

However, as electron transfer reaction in carbon surfaces is slow, various chemical and physical modifications in their surfaces have been proposed to improve the sensitivity, induce selectivity, and eliminate interferences in the determination of organic compounds, including synthetic antioxidants. According to the search performed in this review, the use of a variety of modifications, including nanoparticles, polymers, metals oxides, and ionic liquids, is reported, in which all modifications are performed for improving the sensitivity, selectivity, and robustness of the analysis.

Motia et al. [107] used a screen-printed carbon electrode (SPCE) modified with gold nanoparticles in the preparation of the molecularly imprinted polymer (MIP) sensor in the presence of BHA as templates. This sensor exhibited responses proportional to concentrations over a range of 0.01 to 20 μg/mL, with a low limit of detection of 0.001 μg/mL, with an ability to determine BHA in foodstuff samples (potato chips, mayonnaise, and chewing gum).

Manoranjitham and Narayanan [52] used a voltammetric sensor prepared with multiwall carbon nanotubes (MWCNT) modified by electropolymerized poly O-cresolphthalein complexone in the determination of BHA. After the experimental and voltammetry optimization, the sensor showed a linear range of 0.33 to 110 µmol/L with a limit of detection of 0.11 µmol/L, following the electrochemical determination of BHA in potato chip samples, indicating that the sensor was highly stable and reproducible for BHA determination.

Fan and Kan [108] developed an exfoliated graphite paper electrode modified by silver and nickel oxide nanoparticles for the voltammetric determination of BHT in the presence of its structural analogs and other coexisting substances, resulting in a procedure with a wide linear range of 3.0 × 10^−8^ to 5.0 × 10^−5^ mol/L with a low limit of detection of 2.0 × 10^−8^ mol/L, allowing for the determination of BHA in edible oil samples with satisfactory results.

Additionally, the chemically modified electrodes are suitable in the synthetic phenolic antioxidant and other organic compounds because sometimes the modifications can change the potential in which the reaction of interest occurs, reducing the currents from other chemical compounds that react in the similar potential. Furthermore, the modifications in the working electrode surface can reduce the strong adsorption of the product of the redox reaction, improving the reproducibility in the voltammetric signals.

## 4. Trends and Perspectives in Synthetic Antioxidant Analysis

This review presented some methodologies for the determination of synthetic phenolic antioxidants in foodstuffs, addressing that one of the main factors that ensure the successful determination of these compounds in complex matrices depends a lot on how the antioxidants are extracted from the matrix, as well as the analytical technique used. As chromatographic techniques require extraction and cleaning steps, the determining factor in electrochemical analysis is the surface of the working electrodes where chemical reactions of electron transfers will take place.

In the case of LLE, to bypass the excessive use of organic solvents, aqueous two-phase systems (ATPs) emerged, which are ternary systems that present two phases in thermodynamic equilibrium. In addition to water, the other two that form an ATP can be a combination of polymers, electrolytes, surfactants, or alcohols. Compared with traditional extraction, the constituents of this system are nonflammable and can be biocompatible and biodegradable, in addition to offering advantages, such as short phase separation time, low viscosity, low toxicity, and easy scalability, allowing the efficient extraction of biomolecules, antioxidants, and samples of environmental and food interest [61].

Carbon nanomaterials have proven to be a new class of sorbents primarily due to their large surface area and the ability to modify their surface, both covalently and noncovalently, making them very versatile materials that can interact with a wide variety of compounds. Furthermore, more and more studies have emerged, intending to miniaturize traditional methods to reduce the solvent, in addition to using small amounts of sample with the main objectives of reducing systematic errors and the generation of waste [15,17].

Considering the use of electrochemical techniques and the development and use of working electrodes that can easily be prepared and miniaturized, disposable devices with low cost and without toxic composition are necessary to allow rapid in situ analysis with small volumes of samples in very short times. The possibility of automation in analytical procedures and field analysis with suitable reliability and repeatability is the main perspective. Therefore, the use of screen-printed electrodes, based on the screen-printing technology, presents itself as an interesting possibility, expanding the applicability of voltammetric techniques in quality control and food safety evaluation of foodstuffs. Besides, these electrodes can be easily modified by chemical compounds, including biomolecules and metal nanoparticles, improving their sensitivity and selectivity.

## 5. Conclusions

The employ of chromatographic techniques in antioxidant detection allows suitable sensitivity, selectivity, linear range, and recovery percentages. The modification of the polarity of the mobile and stationary phase improves the separation and permits the detection of a mixture of antioxidants even in complex samples containing high fat, protein, and other additive contents. However, the cost of instrumentation and needs of qualified skilled labor force to operate the instrumentation and perform the steps and preparation of the samples increases the time and cost of analysis, hindering the applicability in quality control in food industries. The development of a simpler instrumentation with the automatization of all steps, including the sample preparation, will provide greater applicability in industries and ease in the development of new research.

The preparation of foodstuff samples and preconcentration of analytes for the analysis is sorely needed. Sample preparation is the main step in foodstuff analysis, and it directly influences the reliability and accuracy of the analysis results. Approaches to green chemistry in sample preparation techniques, as a sustainable and ecologically correct alternative to the classic techniques, are mandatory. At the same time, green sample preparation techniques are fast, simple, generally solvent-free, sensitive, reliable, and cost-effective. Microextraction techniques play an important role in sample preparation due to their inherent advantages over conventional procedures. The main trends in sample extraction techniques are towards simplification and miniaturization of sample preparation and minimization of sample size and the organic solvent used. In the coming years, it is very likely that more ecological techniques for sample preparation will increasingly be applied in food analysis, which is highly desirable.

Actually, the use of electrochemical techniques allows analytical parameters similar to those obtained using chromatographic techniques; however, chemical analysis results in simplified steps of preparation of the samples, using a simple instrumentation, with low costs and fast analysis. However, the use of electrochemical techniques in synthetic phenolic antioxidant analysis requires the use of chemically modified electrodes to improve sensitivity and allows reproductivity in the voltammetric signals. The adequate choices of modification in the working electrodes result in a suitable analysis of synthetic antioxidants, with a simple, fast, low-cost procedure of preparation of the samples before the voltammetric analysis.

## Figures and Tables

**Figure 1 molecules-27-07137-f001:**
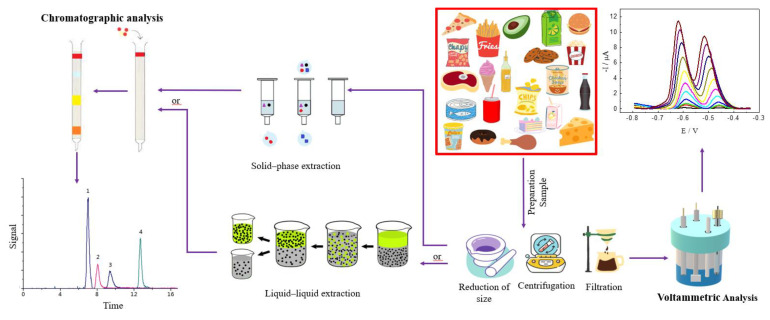
The general scheme employed for the chromatographic and voltammetric analysis of antioxidants indicates the preparation of foodstuff samples in each type of analysis.

**Figure 2 molecules-27-07137-f002:**
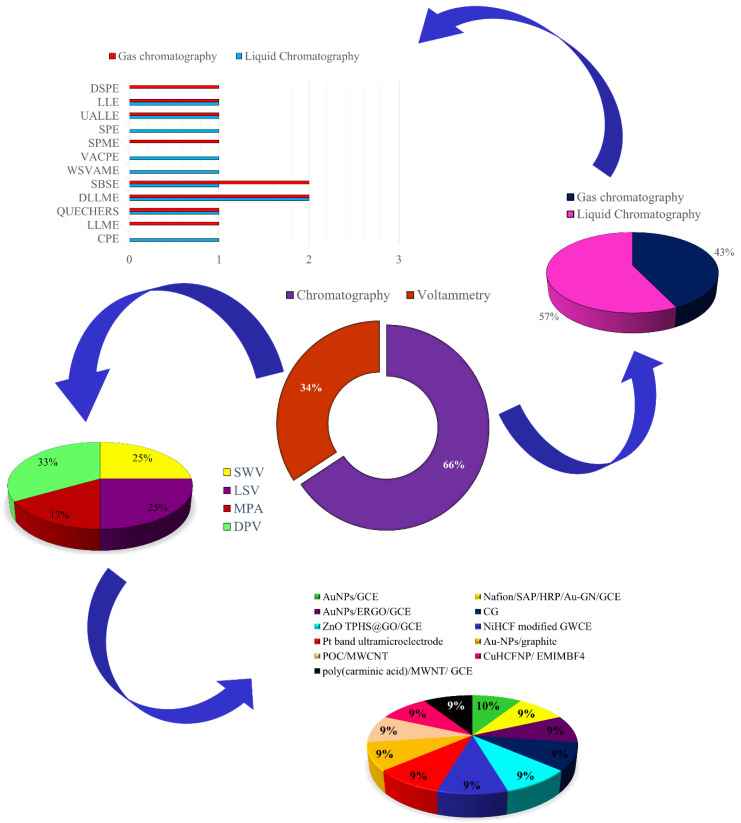
The number of scientific studies published that performed the synthetic phenolic antioxidant analysis using chromatographic detection and respective extraction steps and electrochemical techniques and respective working electrodes, from scientific databases in ScienceDirect^®^ in the years 2011 to 2022. Legend: dispersive solid-phase extraction (DSPE), liquid–liquid extraction (LLE), ultrasonic-assisted liquid–liquid microextraction (UALLME), solid-phase extraction (SPE), solid-phase microextraction (SPME), vortex-assisted cloud-point extraction (VACPE), water-contained surfactant-based vortex-Assisted microextraction (WSVAME), stir bar sorptive extraction (SBSE), dispersive liquid–liquid microextraction (DLLME), quick, easy, cheap, effective, rugged, and safe extraction (QuECHERS), liquid–liquid microextraction (LLME), cloud-point extraction (CPE), square wave voltammetry (SWV), linear scan voltammetry (LSV), multiple-pulse amperometry (MPA), differential pulse voltammetry (DPV). Acronyms in the figure below and to the right are indications of different chemically modified electrodes.

**Table 1 molecules-27-07137-t001:** Main synthetic phenolic antioxidants with their respective molecular formula, chemical structure, acceptable daily intake (ADI), limit-dose 50 (LD_50_), partition coefficient (log K_ow_), and ionization constant (pK_a_). NR: not reported; b.w.: body weight.

Name	Formula	Chemical Structure	ADI (mg/kg by b.w.)	LD_50_ (mg/kg by b.w.)	log k_ow_	pk_a_
**BHA**	C_11_H_16_O_2_	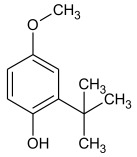 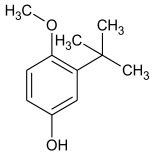	0–0.5	<2000	3.50	8.11
**BHT**	C_15_H_24_O	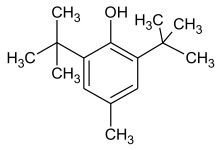	0–0.3	<2000	5.19	12.80
**TBHQ**	C_10_H_14_O_2_	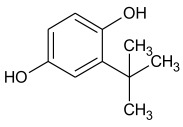	0–0.7	955	2.26	10.80
**PG**	C_10_H_12_O_5_	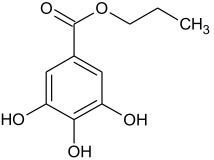	0–1.4	NR	1.78	7.94
**OG**	C_15_H_22_O_5_	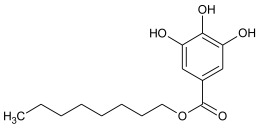	NR	NR	4.33	7.49

## Data Availability

Not applicable.

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
