# Peer review of "Detection of Synthetic Antioxidants: What Factors Affect the Efficiency in the Chromatographic Analysis and in the Electrochemical Analysis?"

_molecules, 2022, doi:10.3390/molecules27207137_

Round 1

Reviewer 1 Report

The authors presented the previous works related to the synthetic antioxidants detection method very well which indicated their respect for readers and their self selfs.
Congratulations to them

To better understand, it is better you put some schemes to explain the process of the detection or the process of the fabrication of the electrochemical sensor. It is also the author mention the disadvantage of the chromatographic analysis and in the voltammetric analysis.

Wish the bests

Author Response

The authors mentioned the disadvantage of the chromatographic analysis and in the electrochemical analysis in the corrected manuscript.

The text was modified to add information about the process of the detection of antioxidants and also an indication of the procedure of working electrode preparation.

Reviewer 2 Report

This work is a comprehensive review of chromatographic and voltammetric methods for the detection of phenolic antioxidants in food products, with focus on the factors that influence detection efficiency.

-The work contains several mentions of amperometric methods of detection. The title refers to "voltammetric analysis". The work should either exclusively refer to voltammetric methods or the title should be rephrased to refer to "electrochemical analysis".

-Table 2 should also include the linear detection range.

-The text between lines 49 and 57 should be made shorter or even eliminated since it is of little relevance to the subject of the review.

-

Author Response

-The title and all manuscript was modified to refer to  "electrochemical analysis" as suggested by the reviewer, since amperometric analysis present important application in the perspectives of chromatographic detector.

Linear range was added in Table 2.

The text between lines 49 and 57 was shortened.

According to Codex Alimentarius [5], there are about 230 compounds classified according to their functionalities in 25 classes of food additives, that are regulated by specific legislation of each country following a rigorous policy of food safety [4,6].